

# Nighttime ozone in the lower boundary layer and its influences on surface ozone: insights from 3-year tower-based measurements in South China and regional air quality modeling

Guowen He[1,2], Cheng He[1,2], Haofan Wang[1,2], Xiao Lu[1,2*], Chenglei Pei[3], Xiaonuan Qiu[3], Chenxi Liu[1,2],
Yiming Wang[1,2], Nanxi Liu[1,2], Jinpu Zhang[3], Lei Lei[3], Yiming Liu[1,2], Haichao Wang[1,2], Tao Deng[4], Qi
Fan[1,2], Shaojia Fan[1,2*]

[1]School of Atmospheric Sciences, Sun Yat-sen University, and Key Laboratory of Tropical Atmosphere-Ocean System,
Ministry of Education, Zhuhai, China
[2]Guangdong Provincial Observation and Research Station for Climate Environment and Air Quality Change in the Pearl River
Estuary, Southern Marine Science and Engineering Guangdong Laboratory (Zhuhai), Zhuhai, China
[3]Guangzhou Sub-branch of Guangdong Ecological and Environmental Monitoring Center, Guangzhou, China
[4]Guangdong Provincial Key Laboratory of Regional Numerical Weather Prediction, Institute of Tropical and Marine
Meteorology, China Meteorological Administration, Guangzhou, China

*Correspondence to*: Xiao Lu (luxiao25@mail.sysu.edu.cn) and Shaojia Fan (eesfsj@mail.sysu.edu.cn)

**Abstract.** Nighttime ozone in the lower boundary layer regulates atmospheric chemistry and surface ozone air quality, but our understanding of its vertical structure and impact is largely limited by the extreme sparsity of direct measurements. Here we present 3-year (2017-2019) measurements of ozone in the lower boundary layer (up to 500 m) from the Canton Tower at Guangzhou, the core megacity in South China, and interpret the measurements with a one-month high-resolution chemical simulation from the Community Multiscale Air Quality (CMAQ) model. Measurements are available at 10 m, 118 m, 168 m,

and 488 m, with the highest 488 m measurement platform higher than the typical height of nighttime stable boundary layer that allows direct measurements of ozone in the nighttime residual layer (RL). We find that ozone increases with altitude in the lower boundary layer throughout the day, with nighttime (daytime) ozone at the 488 m height being 2.4-5.4 (1.5-2.4) times as that at the 10 m height. This indicates a persistent high ozone level and oxidation capacity aloft the surface. The ozone vertical gradient between the 10 m and 488 m height ($\Delta O_3/\Delta H_{10-488 \text{ m}}$) is 3.6-6.4 ppbv/hm in nighttime and 4.4-5.8 ppbv/hm

daytime. We identify a strong ozone residual capacity, defined as the ratio of the ozone concentration averaged over nighttime to that in the afternoon (14:00-17:00 LT), of 67%-90% in January, April and October, remarkably higher than that in the other three layers (29%-51%). Ozone in the afternoon convective mixing layer provides the source of ozone in the RL, and strong temperature inversion facilitates the ability of RL to store ozone from the daytime convective mixing layer, by constraining the exchange of RL ozone with ozone inside the nocturnal stable boundary layer that is subject to strong chemical destruction

and deposition. The tower-based measurement also indicates that nighttime surface $O_x$ ($O_x=O_3+NO_2$) level can be an effective indicator of RL ozone if direct measurement is not available. We further find significant influences of nocturnal RL ozone on both nighttime and the following day's daytime surface ozone air quality. During the surface nighttime ozone enhancement (NOE) event, we observe significant decrease in ozone and increase in $NO_2$ and CO at the 488 m height, in contrast to their



changes at the surface, a typical feature of enhanced vertical mixing. The enhanced vertical mixing leads to NOE event by

introducing ozone-rich air in the RL to enter the nighttime stable boundary layer and weakens the titration effect by diluting

$NO_x$ concentrations. The CMAQ model simulations also demonstrate enhanced positive contribution of vertical diffusion

($\Delta$VDIF) to ozone at the 10 m and 118 m and negative contribution at the 168 m and 488 m during the NOE event. We also

observe strong correlation between nighttime RL ozone and the following day's surface MDA8 ozone. This is tied to enhanced

vertical mixing with the collapse of nighttime RL and the development of convective mixing layer, which is supported by the

CMAQ simulated increase in positive $\Delta$VDIF of +50 ppbv·hr$^{-1}$ at the 10 m and negative $\Delta$VDIF of -10 ppbv·hr$^{-1}$ at 488 m at

early morning (08:00-09:00 LT), suggesting that the mixing of ozone-rich air from nighttime RL downward to surface via the

entrainment is an important mechanism to aggravate ozone pollution in the following day. We find that the bias of CMAQ

simulated surface MDA8 ozone in the following day shows a strong correlation coefficient ($r$=0.74) with the bias in nighttime

ozone in the RL, highlighting the necessity to correct air quality model bias in the nighttime RL ozone for accurate prediction

of daytime ozone. Our study thus highlights the value of long-term tower-based measurements for understanding the coupling

between nighttime ozone in the RL, surface ozone air quality, and boundary layer dynamics.

## 1 Introduction

Ozone is a chemically and radiatively active species affecting atmospheric oxidation capacity and climate, and also harms

human health and ecosystem (Monks et al., 2015; Fleming et al., 2018; Unger et al., 2020). It is generated from nitrogen oxides

($NO_x$=$NO$+$NO_2$), carbon monoxide (CO), and volatile organic compounds (VOCs) in the presence of sunlight, removed by

chemical loss and dry deposition, and can transport at different timescales. Ozone at different vertical layers has distinct

differences in the magnitude of these budget terms, and exchanges through vertical mixing, shaping the highly variable ozone

vertical structure in the boundary layer throughout the day (Tang et al., 2017). Ozone concentrations at the surface (ground-

level) are low during nighttime due to chemical destruction and dry deposition. In the nocturnal residual layer (RL) that is a

few hundred meters above the surface, however, a large amount of ozone may maintain from daytime convective mixing layer

(Mathur et al., 2018; Xu et al., 2018). The lack of direct measurement of nighttime ozone in the lower boundary layer (including

stable boundary layer and the RL) has been limiting our understanding of its role in regulating nighttime chemistry and air

quality. Here we report observed ozone aloft within the lower boundary layer (up to 500m) with a focus on nighttime ozone

and examine its interaction with surface ozone in Guangzhou, the core megacity in South China with severe ozone pollution,

based on hourly measurements at four vertical layers at the Canton Tower for 2017-2019 and chemical simulation from a

regional air quality model.

Nighttime ozone in the boundary layer plays an important role in the atmospheric oxidation capacity (Brown and Stutz, 2012;

Wang et al., 2023) and surface ozone air quality (Klein et al., 2014), and its evolution is highly coupled to the boundary layer

dynamics. After sunset, the surface air decouples from the air aloft with radiative cooling, forming a nighttime stable boundary



layer near surface and a RL from the erstwhile convective mixing layer (Stull, 1988). Due to weak chemical destruction and dry deposition, the RL could largely store ozone concentrations from daytime convective mixing layer, serving as an ozone reservoir (Hu et al., 2018; Caputi et al., 2019; Ouyang et al., 2022). High ozone in the nighttime RL exerts significant influences on the nighttime heterogeneous chemistry (Brown et al., 2016), including the formation of secondary aerosol (Prabhakar et al.,
2017; Yang et al., 2022). Nighttime ozone in the RL also significantly affects surface ozone concentrations. In nighttime, the ozone-rich air in the RL may mix down to surface and trigger a nocturnal ozone enhancement (NOE) event in favorable weather conditions such as the nocturnal low-level jets (Sullivan et al., 2017; He et al., 2022a; Wu et al., 2022). After sunrise, vertical mixing strengthens with increasing ground temperature and the formation of convective mixing layer, accelerating the exchange of ozone-rich air in the RL with air at the surface (Doran et al., 2003; Zhao et al., 2019; Yu et al., 2020). A number
of studies estimated that more than 50% of surface ozone comes from RL in the morning (Lin et al., 2010; Zhu et al., 2020a; He et al., 2021), indicating a significant contribution to daytime ozone air quality from the nighttime ozone in the RL.

Despite the importance of nighttime ozone in the RL in atmospheric chemistry and air quality, direct observations of nighttime ozone in the boundary layer are extremely sparse. So far, existing approaches of ozone vertical profiling include unmanned
aerial vehicle (Wu et al., 2021; Qu et al., 2022), aircraft (Petzold et al., 2015), ozonesonde (Tarasick et al., 2019), ground-based or vehicle-based lidar (Klein et al., 2017; He et al., 2022b), tethered balloon (Zhang et al., 2019), and satellite (Ziemke et al., 2019). The routine ozonesonde and satellite measurements are typically scheduled to observe ozone in the afternoon, therefore nighttime ozone profiles are missing. The unmanned aerial vehicle, aircraft, and tethered balloon measurements of ozone profiles are operated in a low frequency and are dependent on weather conditions. Lidar measurements have the
capability to achieve high spatiotemporal resolution of ozone profiles, however, the blind zone height and the overlap effect of the lidar system invalidate the measurements below hundreds of meters (Lin et al., 2021; Wang et al., 2021). The ozone retrieval algorithm brings additional uncertainty for both satellite and lidar measurements. In comparison, tower-based measurements of ozone and related precursors are unique platforms to investigate the interactions between surface ozone and that aloft in the lower boundary layer throughout the day. It has advantages over other approaches by providing continuous,
simultaneous, and accurate measurements of ozone at different layers in the lower boundary layer in any weather conditions. Tower-based ozone studies in China have been reported with a focus on vertical characteristics and seasonal/decadal changes of ozone and aerosol components (Sun et al., 2010; Li et al., 2019; Li et al., 2022a; Li et al., 2022b; Liu et al., 2022b). However, much less attention has been attached to the ozone evolution in nighttime in the RL and its interaction with surface ozone.

Here, we present 3-year (2017-2019) measurements of lower boundary layer ozone from the 610 m height Canton Tower in Guangzhou, the core megacity in South China where ozone concentrations are high and have been increasing (Wang et al., 2019b; Li et al., 2020; Lu et al., 2020) even with controls on anthropogenic $NO_x$ emissions (Zheng et al., 2018; Zhong et al., 2018). We interpret the observations with a one-month high-resolution chemical simulation from the Community Multiscale Air Quality (CMAQ) model. We analyze the observed nighttime versus daytime vertical structure and quantify ozone vertical



gradient in different seasons. We then investigate the chemical and meteorological factors controlling the nighttime ozone in the RL from the measurement at the 488 m height, which is typically higher than the nighttime stable boundary layer and can be representative of RL. We further examine the roles of nighttime RL ozone in both nocturnal ozone enhancement events and the following day's surface ozone air quality.

## 2 Data and methods

### 2.1 Observational data

We obtain hourly measurements of ozone, NO$_2$, CO, and meteorology parameters (temperature, wind speed) at the Canton Tower, the second highest TV tower in the world with a total height of 610 m including the shaft on the top. The Canton tower (23.11°N, 113.33°E) is located in the center of urban Guangzhou, the core city in South China and a typical city influenced by the East Asia monsoon (Fig. 1). Measurements are conducted by the Guangzhou Environmental Monitoring Center. The

instruments are arranged in open area of the Canton Tower, therefore the measurement can reflect the property of the ambient air.

Ozone is measured online hourly via the Thermo 49i instrument placed in the tower at four vertical layers (10 m, 118 m, 168 m, and 488 m), extending from the surface to the lower boundary layer. Simultaneous measurement of temperature (10 m and 118 m), wind speed (10 m, 118 m, 168 m, and 488 m), NO$_2$, and CO (10 m, 118 m, 168 m, and 488 m) are available.

Measurements are available in 2017-2019 for January, April, and October, but are only available in 2019 for July due to instrument malfunction in other summers. Previous studies have illustrated the Canton Tower as a unique platform to study the vertical distribution, chemical reactivity, and source attribution of ozone and other pollutants (Mo et al., 2020; Zhou et al., 2020; Li et al., 2022b; Mo et al., 2022; Yang et al., 2022). These studies typically cover a short slice of time periods (from a

few days to a few months) and do not focus on nighttime ozone structure and controlling factors.

We additionally obtain vertical ozone profiles in the lower boundary layer from the nearby ozonesonde station in Hong Kong and reported tower-based observations at other cities in mainland China, in order to compare the ozone vertical profiles from the Canton Tower with other measurements. Ozonesonde observations from Hong Kong Observatory are publicly available at

the World Ozone and Ultraviolet Radiation Data Center (https://woudc.org/data/explore.php?lang=en, last access: 06 April 2023). Each sounding is launched on a weekly basis at local time 13:00–14:00 at the King's Park Meteorological Station (114.17°E, 22.31°N, 66 m above sea level). A total of 45 profiles from 2017-2019 are obtained in this study (12 in January, 10 in April, 11 in July, and 12 in October, respectively). We extract the ozone concentrations from each profile at 6 levels (the lowest detection height, 100 m, 200 m, 300 m, 400 m, and 500 m). We also collect reported observations from the Beijing

Meteorology Tower (325 m) (Liu et al., 2022b), Tianjin Meteorological Tower (250 m) (Han et al., 2020), Shenzhen Meteorology Tower (356 m) (Li et al., 2019), and another independent study at the Canton Tower (Li et al., 2022b). However,



these studies do not separately report nighttime measurements, thus we focus the comparison on mean vertical ozone distributions.

## 2.2 Reanalysis data of planet boundary layer height

We also obtain the planet boundary layer height (PBLH) from reanalysis dataset to describe the nighttime and daytime PBLH at the Canton Tower. Previous studies have evaluated the PBLH from several widely-used reanalysis products, including the fifth-generation ECMWF (European Centre for Medium-Range Weather Forecasts) atmospheric reanalysis of the global climate (ERA5), the Modern-Era Retrospective Analysis for Research and Applications, version 2 (MERRA-2), the National Centers for Environmental Prediction final global reanalysis data (FNL), and the coupled ECMWF atmospheric reanalysis of

medium-range weather (CERA) across China (Guo et al., 2021; Xu et al., 2022), with radiosounding observations. These studies highlight the robustness and suitability of PBLH estimate from the ERA5 reanalysis dataset. The ERA5 is the fifth generation ECMWF atmospheric reanalysis of the global climate, which is generated by assimilating radiosonde and several satellite observational datasets ([https://cds.climate.copernicus.eu/#!/search?_text=ERA5](https://cds.climate.copernicus.eu/#!/search?_text=ERA5), last access: 08 April 2023). The ERA5 dataset has a horizontal resolution of 0.25°×0.25° and a vertical resolution of 37 pressure levels with 1-hour temporal

resolution, and is able to describe the diurnal characteristics of boundary layer. We also apply the global continental blended high-resolution PBLH dataset developed by Guo. et al. (2022) ([https://zenodo.org/record/6498004](https://zenodo.org/record/6498004), last access: 11 April 2023), which is generated by integrating ERA5 reanalysis dataset, the Global Land Data Assimilation (GLDAS) System, and measurements at over 370 radiosonde stations worldwide using machine learning algorithms. The dataset has a same horizon resolution of 0.25°×0.25° as the ERA5 dataset but is only available every 3 hours.


## 2.3 WRF-CMAQ model simulation

We use the CMAQv5.4 model to interpret ozone measurements at the Canton Tower and to diagnose the interactions between surface ozone and that aloft in the lower boundary layer. The CMAQ model is a third-generation state-of-art air quality modeling system developed by the United States Environmental Protection Agency (US EPA) (Appel et al., 2021). The model

is driven by meteorological input data simulated from the Weather Research and Forecasting model (WRF) version 3.9.1 (Skamarock et al., 2008). The initial and boundary conditions for the WRF model are obtained from the National Center for Environmental Prediction (NCEP) FNL dataset with a horizontal resolution of 1°×1° and a temporal resolution of 6-hour.

Table 1 provides a summary of the chemical and physical schemes, as well as emission inventories used in the WRF-CMAQ

model. The key configurations of WRF-CMAQ model include the Radioactive Transfer Model (RRTM) for short and long wave radiation scheme, the Noah land surface model (LSM) for land-atmospheric interactions, the MYJ boundary layer scheme, the Purdue Lin microphysics scheme, the Carbon Bond 06 (CB06) combined with AERO6 for gas-phase and aerosol chemistry. Anthropogenic emissions in China are from the Multiscale Emission Inventory of China (MEIC) (Zheng et al., 2018). Biogenic



emissions were calculated by the Model of Emissions of Gases and Aerosols from Nature version2.1 (MEGAN v2.1) (Guenther
et al., 2012).

We set up three nested domains of the WRF-CMAQ model at a horizontal resolution of 27×27, 9×9, and 3×3 km², respectively,
with the innermost domain focusing on the PRD region (Fig. 1). For each domain we set up 40 vertical layers with the near-
surface layer at about 20 m above the ground. We conduct a one-month simulation focusing on October 2017 when high ozone
concentrations are observed in Guangzhou and the anthropogenic emission inventory is available. The model simulation starts
from September 20, 2017 to November 1, 2017 with the first 10 days used as spin-ups.

We apply the integrated process rate (IPR) diagnostic module implemented in the CMAQ model to quantify the physical and
chemical influences on ozone budget (Wang et al., 2019a; Ouyang et al., 2022). The IPR diagnoses the change of ozone at
each model grid from vertical advection (ΔZADV), horizontal advection (ΔHADV), horizontal diffusion (ΔHDIF), vertical
diffusion (ΔVDIF), dry deposition (ΔDDEP), gas-phase chemistry (ΔCHEM) and aerosol process (ΔAERO).

## 3 Results and discussions

### 3.1 Mean nighttime versus daytime ozone vertical structure and budget in the lower boundary layer

Figure 2 shows the 2017-2019 monthly mean ozone diurnal cycle in the lower boundary layer measured at the Canton Tower
in different seasons. Higher monthly mean ozone concentrations of 33 ppbv at the 10 m and 57 ppbv at the 488 m height are
observed in October than other seasons. This seasonal ozone cycle is different from that in the northern mid-latitudes which
typically shows ozone peak in boreal summer (Lu et al., 2019b), reflecting the dry and hot weather conditions in October in
Southern China that are favorable for ozone chemical production with the retreat of summer monsoon (Yin et al., 2019; Gao
et al., 2020). Despite the difference in mean values, the diurnal cycle is consistent throughout the lower boundary layer from
10 m to 488 m above the ground and for all seasons, with ozone increasing after sunrise, peaking in afternoon, and decreasing
gradually after sunset. Slight difference lies in the time of ozone increase after sunrise between different vertical layers. The
lowest three layers (10, 118, 168 m) show ozone increase at about 8 am, while ozone in the highest 488 m layer typically
shows a small decrease at 7-9 am and starts to increase after 10 am.

We see from Figure 2 that ozone concentrations are generally higher with increasing altitudes in the lower boundary layers.
Figure 3 presents the 2017-2019 mean vertical ozone profile that reinforces this feature. The vertical ozone structure observed
at the Canton Tower is generally consistent with that obtained from the nearby ozonesonde in Hongkong and other reported
tower-based measurements in China (Fig. S1), however, our analyses here provide additional information on the comparison
of ozone structures in daytime (08:00-19:00 LT) and nighttime (20:00-07:00 LT). In nighttime (daytime), the monthly mean
ozone concentrations at the 488 m are 38±18 (42±20) (mean ± standard deviation from hourly measurements), 37±16 (48±19),



23±7 (47±15), and 50±19 (64±21) ppbv in January, April, July, and October, respectively (Table. 2). In comparison, ozone concentrations at 10 m above the surface in nighttime (daytime) are 7±7 (17±12), 8±7 (25±16), 6±6 (20±15), and 20±12 (43±20) ppbv for the corresponding months. This shapes a significant vertical gradient of ozone concentrations between the lower boundary layer and the surface layers, with nighttime (daytime) ozone concentrations higher by 2.4-5.4 (1.5-2.4) times at 488 m level than that at the surface.

We express the observed monthly-mean vertical gradient of ozone as $\Delta O_3/\Delta H$ (ppbv/hundred-meters (hm)) derived from observation platforms at different heights. The overall ozone vertical gradient in the lower boundary layer ($\Delta O_3/\Delta H_{10\text{-}488\ m}$) are 6.4, 6.1, 3.6, and 6.1 ppbv/hm in nighttime, compared to 5.2, 4.9, 5.8, and 4.4 in daytime, suggesting a larger vertical ozone gradient between 488 m and 10 m level in the nighttime than daytime. However, the daytime versus nighttime $\Delta O_3/\Delta H$ features vary for different altitude range (Table. 3). In the lowest layer of 10-118 m above the ground, the ozone vertical gradient is typically smaller in nighttime, with $\Delta O_3/\Delta H_{10\text{-}118\ m}$ ranging from 3.2-8.6 ppbv/hm in nighttime versus 5.8-27.6 ppbv/hm in daytime. In the higher layer of 168-488 m, however, nighttime $\Delta O_3/\Delta H_{168\text{-}488\ m}$ values are 2.9-7.1 ppbv/hm, higher than the -0.6-4.7 ppbv/hm in daytime.

The larger nighttime ozone vertical gradient than daytime vertical gradient reflects the evolution of atmospheric boundary layer. We analyze the boundary layer height at the Canton Tower derived from ERA5 reanalysis dataset and the global continental blended high-resolution planetary boundary layer height (PBLH) dataset developed by Guo. et al. (2022), as described in Section 2.2. The two datasets are consistent in the boundary layer height and its diurnal variation (Fig. S2). In daytime, we find that mean PBLH is 527±306, 694±372, 716±319, and 722±417 m in January, April, July, and October, respectively. The highest (488 m) measurement platform of the Canton Tower is in the convective mixing layer, in which ozone vertical gradient is relatively small as a result of rapid vertical mixing. In comparison, in nighttime, the average PBLH (denotes the height of nighttime stable boundary layer) are 221±185, 253±176, 263±177, and 215±188 m for the corresponding months, with 85%-89% of the hourly records showing PBLH smaller than 488 m, indicating that the 488 m platform is able to observe ozone in the RL. Therefore, nighttime ozone concentrations at 10, 118, and 168 m are relatively close to each other, while ozone concentrations at 488 m in the RL remain high and are decoupled from the lowest three layers in the nighttime stable boundary layer. This also corresponds to a decreasing day-to-night ozone contrast (difference between maximum and minimum ozone concentrations within the day) decreases with increasing height except for July as shown in Figure 2. At the surface, the day-to-night ozone contrast is 25, 32, 25, and 45 ppbv in January, April, July, and October, respectively, compared to 16, 26, 48, and 33 ppbv at the 488 m height, reflecting a smaller ozone diurnal cycle in the RL compared to the lower altitudes.

We use the one-month CMAQ model simulation of October 2017 to diagnose the processes controlling the ozone evolution of lower boundary layer in different altitudes. Evaluation of the simulated meteorological parameters and ozone concentrations



with measurements at the Canton Tower is summarized in Table S1. The CMAQ model captures the daily variation and the
diurnal cycle of the observed ozone concentrations, with the correlation coefficient (*r*) of 0.70 and the mean bias (MB) of -1.8
ppbv for all 744 hourly data points (Fig. 4). However, the model tends to overestimate daytime surface ozone concentrations
and underestimate nighttime surface ozone concentrations. This is a common feature for most current state-of-art chemical
transport models. Even with a resolution of 3 km, there is still representative issue when comparing gridded model results to

ozone measurements at a specific site. Artificial mixing of ozone precursors in model grids tend to overestimate ozone chemical
production efficiency in daytime and the titration effect in nighttime, which leads to an overestimation of daytime ozone and
underestimation of nighttime ozone (Lu et al., 2019b). Limitations in chemical mechanism, emission inventories, and modeling
fine-scale meteorology bring extra uncertainty in high-resolution regional ozone simulations (Liu and Wang, 2020; Liu et al.,
2022a; Yang and Zhao, 2023). Nevertheless, the model largely reproduces the nighttime vertical ozone gradient $\Delta O_3/\Delta H_{10\text{-}488}$

$_m$ of 5.1 ppbv/hm, compared to 6.0 ppbv/hm from the observations in October 2017, though it has difficulty to capture the
magnitude of daytime vertical ozone gradient (Fig. 4).

Figure 4 summarizes the factors controlling daytime versus nighttime ozone budgets at different heights, derived from the
CMAQ IPR diagnostic module. We find that vertical diffusion ($\Delta$VDIF) is the main source of ozone at the surface, while net

chemical loss ($\Delta$CHEM) due to the net effect of ozone chemical production and destruction is the main loss. Dry deposition
($\Delta$DDEP) also contributes to ozone decrease in the surface. In daytime, $\Delta$VDIF contributes to +70 and -6 ppbv·hr$^{-1}$ of ozone
change rate at the surface and 488 m, respectively, while in nighttime, the corresponding contributions are +35 and -2 ppbv·hr$^{-1}$.
This implies that $\Delta$VDIF has different contribution to ozone budget between the surface (source) and in the lower boundary
layer (sink), and that there is a strong impact of $\Delta$VDIF on surface ozone exists in both daytime and nighttime. $\Delta$CHEM

contributes +2 ppbv·hr$^{-1}$ in daytime and -1 ppbv·hr$^{-1}$ in nighttime to ozone change rate at 488 m, respectively, indicating that
chemistry is not a major source/sink of ozone at the height of 488 m.

## 3.2 Nighttime ozone in the residual layer and affecting factors

Our analyses above show that the mean height of nighttime stable boundary layer at the Canton Tower is typically lower than

488 m, allowing us to use measurements at the 488 m height from the Canton Tower to probe into ozone level in nighttime
RL. This represents a major advantage of the Canton Tower for analyzing nighttime ozone evolution in both the stable
boundary layer and RL, compared with other existing tower measurements in other parts of China with a limited observation
height of less than 400 m (Qiu et al., 2019; Li et al., 2022a; Liu et al., 2022b). We take advantage of this to examine RL's
ability to serve as a reservoir of ozone and its influencing factors.


We examine in Figure 5 the nighttime ozone residual capacity at different heights, defined as the ratio of the ozone
concentrations averaged over nighttime to that averaged over afternoon (14:00-17:00 LT), when ozone concentrations typically



reach the daily peak and the vertical mixing is expected to be the strongest. As shown in Figure 5, the nighttime ozone residual capacity increases with altitude. In particular, the residual capacity at 488 m ranges from 43% in July to 90% in January,
remarkably higher than that in other three layers of 29%-51%. It indicates that except for July, ozone at the 488 m height during nighttime can, in average, reach about 70% or above of the afternoon ozone level. The much larger nighttime ozone residual capacity between the layers also suggests that nighttime ozone at 488 m is well decoupled from the other three vertical layers. It is consistent with our analyses from PBLH that measurement at 488 m can be representative of RL.

We then analyze the factors affecting nighttime ozone level in the RL. The high ozone residual capacity at 488 m as shown in Figure 5 suggests ozone concentrations in the afternoon largely determine nighttime ozone in the RL. Figure 6 shows a strong dependence of nighttime ozone concentrations at 488 m on that in the afternoon, with the correlation coefficients ranging from 0.53 to 0.67 in different months. Such strong correlation between the nighttime and afternoon ozone concentration is not observed at the 10 m measurement, reflecting the different ozone budgets in nighttime stable boundary layer and the RL. We
also see that both high daytime and nighttime ozone concentrations are linked to high PBLH. High PBLH is typically coincident with high air temperature that favors ozone chemical production in $NO_x$-rich environment (Fu and Tian, 2019; Lu et al., 2019a), and also features vigorous atmosphere turbulence that enhances the vertical mixing of ozone (Dai et al., 2023), allowing ozone-rich air at higher altitudes to mix with air in the lower boundary layer.

High ozone in the afternoon convective mixing layer provides the source of nighttime ozone in the RL, but whether ozone can be reserved in the RL also depends on nighttime mixing of air between the RL and the stable boundary layer. We then investigate the dependence of nighttime 488 m ozone concentrations on the concurrent temperature vertical lapse rate γ (ΔT/Δz), an indicator of atmospheric stability, derived from the temperature measurement between the 10 m and 118 m heights. Figure 7 illustrates the clear positive dependence of the 488 m ozone concentrations on γ in nighttime for all seasons. In the presence
of the temperature inversion (γ>0), the mean nighttime 488 m ozone concentrations can reach to 52-68 ppbv in January, April, and October, significantly higher than ozone concentrations with γ<0 of 32-52 ppbv.

We further compare nighttime vertical ozone profiles with γ < 0 (non-inversion) versus that with γ > 0 (inversion) in Figure 7. Here we rule out cases with wind speeds above 2 m/s at four vertical layers, to exclude the possible influence of horizontal
transport. We find significantly a steeper ozone vertical gradient when atmospheric inversion occurs (γ > 0) with the $\Delta O_3/\Delta H_{10\text{-}488\,m}$ ranging from 7.6-12.9 ppbv/hm compared to 3.5-6.1 ppbv/hm in the absence of temperature inversion. The presence of inversion also leads to a much lower nighttime stable boundary layer of 98±72 m compared to non-inversion cases, leading to a strong ozone gradient between 10m and 118m. The atmospheric inversion constrains the exchange of ozone at 488 m and ozone inside the stable boundary layer that are subject to strong chemical destruction and deposition, thus allowing high ozone
concentrations to remain in the RL.



Previous studies suggest using nighttime $O_x$ ($O_x=O_3+NO_2$) concentrations to indicate nighttime ozone in the RL (Wang et al., 2018; Tan et al., 2021; He et al., 2022a), based on the rapid nighttime chemical conversion from ozone to $NO_2$ ($O_3+NO \rightarrow NO_2+O_2$) at surface. Our long-term tower-based observations can be applied to evaluate this assumption. We find

in Figure 8 strong correlation coefficients between nighttime 488 m ozone and nighttime surface $O_x$ of 0.76, 0.73, and 0.85 in January, April, and October, respectively, except for July with $r=0.23$ that may be affected by the small data samples. The slope of nighttime 488 m ozone to nighttime surface $O_x$ are 0.67, 0.91, and 0.90 in January, April, and October, respectively, reflecting influence of nighttime $NO_x$ emissions at the surface. Our analyses above prove that nighttime surface $O_x$ can be a good indicator of nighttime ozone in the RL if there are no direct observations.


## 3.3 Nighttime surface ozone enhancement events linked to enhanced ozone mixing from the residual layer

The direct observations of ozone in nighttime RL allow us to examine its roles in modulating surface ozone air quality. This section discusses the role in the nighttime surface ozone enhancement events. Section 3.4 discusses the role in the following day's ozone air quality.


Previous study has reported high frequency of nighttime ozone enhancement (NOE) events in China (He et al., 2022a). A number of studies suggested that enhanced nighttime vertical mixing induced by specific synoptic processes such as convective storms or low-level jets was a key driver for the NOE events, by allowing the ozone-rich air in the RL to mix with the surface layer ozone. These studies were mainly based on surface measurements, while the lack of direct measurements in the lower

boundary layer and RL limit the validation of the proposed mechanisms (Zhu et al., 2020b; He et al., 2022a; Wu et al., 2022). Here our 3-year tower-based observation at the Canton Tower provides a unique opportunity to examine the proposed mechanism for such events. We follow previous studies to define an NOE event if surface (10 m) ozone concentration increases by more than 5 ppbv ($\Delta O_3/\Delta t > 5$ ppbv h$^{-1}$) in one of any two adjacent hours during nighttime (Eliasson et al., 2003; Zhu et al., 2020b; He et al., 2022a). For comparison, we also define non-enhanced nocturnal ozone (NNOE) event when the maximum

ozone enhancement in all adjacent hours is less than 1 ppbv (maximum of $\Delta O_3/\Delta t < 1$ ppbv h$^{-1}$). We find 75 NOE events (24%) among all the days in 2017-2019 with available observations. This ratio is generally consistent with a recent study that reported a NOE frequency of 19% in Guangzhou based on 2014-2019 observations at 6 monitoring sites (He et al., 2022a).

Figure 9 shows the mean nighttime evolution during the NOE events at the four different observation heights at the Canton

Tower. We find ozone enhancement of 8 ppbv at the surface (10 m) level and 4 ppbv at the 118 m and 168 m between 00:00 LT and 05:00 LT when averaged over all 75 events. The small magnitude of ozone increase reflects the different occurrence time of NOE events among individual events (Fig. S3), and we see large nighttime ozone enhancement by up to 47 ppbv in specific episode. In comparison, we find that ozone decreases at 488 m level, in contrast to ozone increases at the lower three





levels (Fig. 9a). These features are not shown in NNOE events, when ozone concentrations are decreasing gradually between
18:00 LT and 01:00 LT and remain stable afterward for all vertical layers (Fig. 9b).

Figure 10 further compares the vertical profiles of ozone, $NO_2$, and CO concentrations in the hour before and during the
occurrence of NOE. We find significant ozone enhancement by 12 ppbv at the surface level while decrease by 6 ppbv at 488
m during the hour of NOE occurrence. We also see sharp decreases in the $NO_2$ and CO concentrations at the surface level but
slight increases at 488 m, in contrast to the NNOE events when $NO_2$ and CO concentrations at 488 m show continuous decrease.
These features all point to enhanced vertical mixing in the NOE events. Ozone concentrations are higher in nighttime RL than
at the surface, while $NO_2$ and CO concentrations are much higher at the surface as they are primarily released from
anthropogenic emissions at near surface while there is almost no direct source at higher altitudes. As such, the increase in $NO_2$
and CO at 488 m during the NOE events, even though with very small magnitude, is most likely from the vertical mixing with
surface layer air mass. Thus, the contrast change of ozone, $NO_2$, and CO in the 488 m height versus that at the surface provides
direct observational evident that vertical mixing between the two layers is strengthened during the NOE events. The enhanced
vertical mixing promotes the downward mixing of ozone-rich air from the RL to the surface, and at the meantime decrease
surface $NO_x$ concentrations and weakens the titration, thus contributing to the NOE events.

CMAQ model simulation in October 2017 further supports the abovementioned mechanisms for the NOE events. As shown
in Figure 11, the model simulation suggests that vertical diffusion ($\Delta$VDIF) is the major source of nighttime ozone at near
surface (10 m and 118 m), while it is a strong sink for ozone in the higher level. We find significant difference in the magnitude
of $\Delta$VDIF between the NOE and NNOE. In particular, we see large and increasing positive $\Delta$VDIF during the NOE events at
10 m and 118 m, while negative $\Delta$VDIF are found at the 168 m and 488 m (Fig. 11a). In comparison, the $\Delta$VDIF remain small
during the NNOE events (Fig. 11b).

### 3.4 Significant contribution of nighttime residual layer ozone to the following day's daytime surface ozone

The observed high ozone concentrations in nighttime RL suggest that it can be a critical ozone source for the following day's
surface ozone when the RL collapses with the development of convective mixing layer after sunrise. Figure 12 shows that
surface MDA8 ozone in the following day is positively correlated with the mean 488 m nighttime ozone, with $r$ of 0.67
(January), 0.47 (April), and 0.54 (October), respectively, except for July due to small data samples. We find that when the 488
m nighttime ozone is in the range of 60-80 ppbv, surface MDA8 ozone level in the following day can reach 67 ppbv in average.

The observed high correlation between the mean 488 m nighttime ozone and surface MDA8 ozone in the following day is tied
to the enhanced downward mixing of the ozone-rich air in nighttime RL to the surface in the early morning. Figure 2 has
showed that the observed ozone at 10 m starts to increase in 08:00-09:00 LT, while ozone at 488 m decreases concurrently.



The CMAQ model simulation for October 2017 captures the contrast ozone change between 10 m and 488 m in 08:00-09:00 LT (Fig. 13), and diagnoses a positive contribution of ΔVDIF of about 50 ppbv hour$^{-1}$ at the surface while a negative contribution of about -10 ppbv hour$^{-1}$ at 488 m, which can explain the observed ozone change at the 10 m and 488 m. These analyses support that the mixing of ozone-rich air from nighttime RL downwards to the ozone-poor air at the surface via the entrainment is an important mechanism to aggravate ozone pollution in the following day.

The strong correlation between the nighttime RL ozone and the surface MDA8 ozone suggests that the level of nighttime RL ozone can be used a powerful predictor for surface ozone air quality. It also has important implication to improve the bias in simulating daytime ozone using chemical transport models. Our CMAQ model simulation captures the positive relationship between nighttime ozone in the RL and the surface MDA8 ozone in the following day with $r$ of 0.43. In addition, we find that the bias of simulated surface MDA8 ozone in the following day shows a strong correlation coefficient ($r$=0.74) with the bias in nighttime ozone in the RL to the observation (Fig. 14), suggesting that errors in modeling ozone in nighttime RL will propagate to surface ozone in the following day. This result highlights the importance to improve the model capability in simulating nighttime ozone in the RL, in terms of both chemical mechanisms and boundary layer dynamics, for daytime ozone air quality forecast.

**4 Conclusions**

In this study, we present 3-year (2017-2019) measurements of lower boundary layer ozone (up to 500 m) from the Canton Tower, with a focus on nighttime ozone evolution and the interaction between ozone in the RL and that at the ground level. This takes the advantage of the Canton Tower which provides ozone measurement at a height of 488 m, higher than the typical height of the nighttime stable boundary layer derived from the reanalysis dataset so that direct measurement in the RL becomes available. We combine the Canton Tower measurement and the one-month WRF-CMAQ simulation to examine the nighttime versus daytime ozone vertical gradient, the ability of the RL to serve as the ozone reservoir and its chemical and meteorological influencing factors, and further investigate the RL ozone's impact on both nocturnal ozone enhancement event and following day's surface MDA8 ozone.

Our findings are summarized in Figure 15. We find that ozone at different heights in the lower boundary layer show consistent diurnal and seasonal cycle, with ozone peaking in daytime and in October. In both nighttime and daytime and for all seasons, ozone increases with altitude, with nighttime (daytime) ozone at the 488 m height being 2.4-5.4 (1.5-2.4) times as that at the 10 m height, suggesting a persistent high ozone level and oxidation capacity aloft the lower boundary layer throughout the day. We quantify an ozone vertical gradient between the 10 m and 488 m height ($\Delta O_3/\Delta H_{10\text{-}488 \text{ m}}$) of 3.6-6.4 ppbv/hm in nighttime and 4.4-5.8 ppbv/hm daytime, averaged over 2017-2019, reflecting much weakened vertical mixing in the nighttime than



daytime. WRF-CMAQ model diagnoses that vertical diffusion (ΔVDIF) has different contribution to ozone budget between the surface (source) and in the lower boundary layer (sink).


We identify a strong ozone residual capacity (the ratio of nighttime to afternoon ozone) in the RL, ranging 67%-90% in January, April, and October, remarkably higher than that in the other three layers (29%-51%). Ozone in the afternoon convective mixing layer provides the source of ozone in the RL, and strong temperature inversion facilitates the ability of RL to store ozone by constraining the exchange of RL ozone with ozone inside the stable boundary layer that are subject to strong chemical

destruction and deposition. The tower-based measurement also indicates a significant positive relationship ($r$=0.73-0.85) between nighttime ozone at the 488 m height with nighttime $O_x$ ($O_x$=$O_3$+$NO_2$) at the surface (10 m), and a slope of 0.67-0.90 for January, April, and October between the two variables, suggesting that surface $O_x$ level can an effective indicator of RL ozone if direct measurement is not available.

We further find significant influences of nocturnal RL ozone on both nighttime and the following day's daytime surface ozone air quality. In nighttime, ozone mixing from the RL can trigger the nighttime ozone enhancement event (NOE). This is supported by the observed contrast in the ozone, $NO_2$, and CO changes at the 488 m height and at the surface during the NOE event, and also by CMAQ model simulation. The measurements show a significant ozone decrease at the 488 m level (6 ppbv in average) but increase (12 ppbv in average) at the surface during the occurrence of NOE. There are also slight increases in

$NO_2$ and CO concentrations at 488 m, where no direct source of $NO_2$ and CO is expected, as such the increase is more likely due to mixing with air in the nighttime stable boundary layer. The enhanced downward mixing allows ozone-rich air in the RL to enter the nighttime stable boundary layer and also weakens the titration by diluting NOx concentrations, together contributes to the NOE event. The CMAQ model simulations also demonstrate the enhanced positive contribution of ΔVDIF to ozone at the 10 m and 118 m and the negative contribution at the 168 m and 488 m during the NOE event.


In the following day, both observations and CMAQ modeling suggest a decrease in 488 m ozone at early morning (08:00-09:00 LT) with the collapse of nighttime RL and the development of convective mixing layer, and the model diagnoses a significant increase in ΔVDIF of (50 ppbv·hr$^{-1}$) at the 10 m while a negative ΔVDIF of (-10 ppbv·hr$^{-1}$) at 488 m. This partly contributes to the observed positive correlation between the nighttime RL ozone and the following day's surface MDA8 ozone,

suggesting that the mixing of ozone-rich air from nighttime RL downward to the ozone-poor air at the surface via the entrainment is an important mechanism to aggravate ozone pollution in the following day. It also implies that that the level of nighttime RL ozone can be used a powerful predictor for surface ozone air quality. We further find that the bias of CMAQ simulated surface MDA8 ozone in the following day shows a strong correlation coefficient ($r$=0.74) with the bias in nighttime ozone in the RL to the observation. Therefore, correcting air quality model bias in the nighttime RL ozone can be important

for daytime ozone air quality forecast.



Our study thus illustrates the value of long-term tower-based measurements for understanding the coupling between nighttime ozone in the RL, surface ozone air quality, and boundary layer dynamics. Nevertheless, our study mainly focuses on the mean structure and evolution of nighttime ozone in the lower boundary layer on the basis of three-year measurements, and does not

zoom in specific episodes in which the interactions can be more significant and complex. We also call for expanded, continuous, and simultaneous vertical measurements of ozone and other atmospheric components, including VOCs, radicals, and aerosol components, to improve our understanding on the nighttime atmospheric chemistry from the surface to the boundary layer and to better constrain air quality models.

**Data availability.** The ozonesonde data are from World Ozone and Ultraviolet Radiation Data Centre (https://woudc.org/data.php, last access: 06 April 2023). The ERA5 reanalysis data can be accessed via https://cds.climate.copernicus.eu/#!/search?_text=ERA5 (last access: 08 April 2023). The global continental blended high-resolution PBLH dataset is obtained from https://zenodo.org/record/6498004 (last access: 11 April 2023).

**Author contributions.** XL and SJF designed the study. GWH, CH, and HFW conducted the WRF-CMAQ model simulation and analyses with contributions from CLP, XNQ, CXL, YMW, NXL, JPZ, LL, YML, HCW, TD, and QF. CLP, XNQ, JPZ, and LL contributed to measurement. All authors provided practical comments. GWH, XL, and SJF wrote the paper with input from all co-authors.

**Competing interests.** The contact author has declared that none of the authors has any competing interests

**Financial support.** This research has been supported by the Key-Area Research and Development Program of Guangdong Province (grant no. 2020B1111360003), the Guangdong Basic and Applied Basic Research project (grant no. 2020B0301030004), the National Natural Science Foundation of China (grant no. 42105103), and the Guangdong Basic and
Applied Basic Research Foundation (2022A1515011554).



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





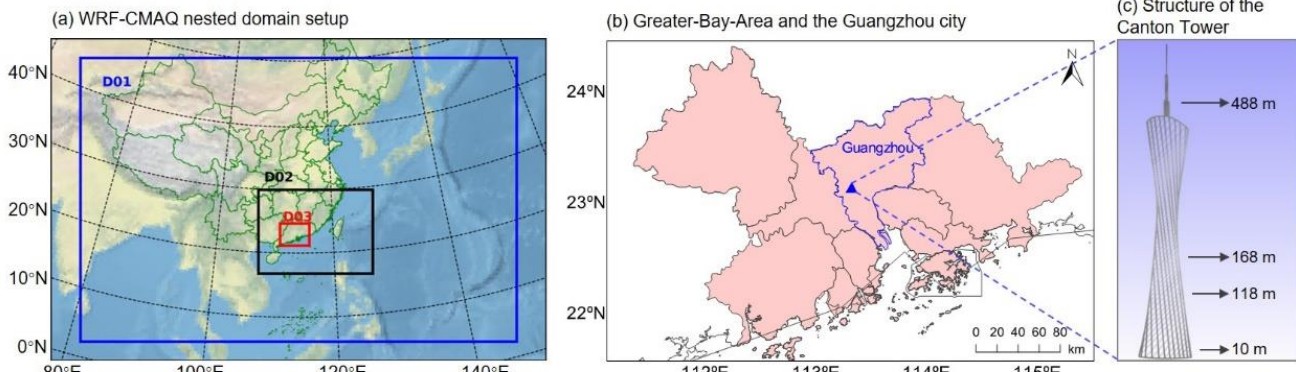

**Figure 1.** Location of the Canton Tower and the region of interest in this study. Panel (a) shows the three nested domains of the WRF-CMAQ model. Panel (b) shows the Greater-Bay-Area and the Guangzhou city. Panel (c) shows the structure of the
685 Canton Tower, with the four measurements heights (10 m, 118 m, 168 m, and 488 m) denoted.




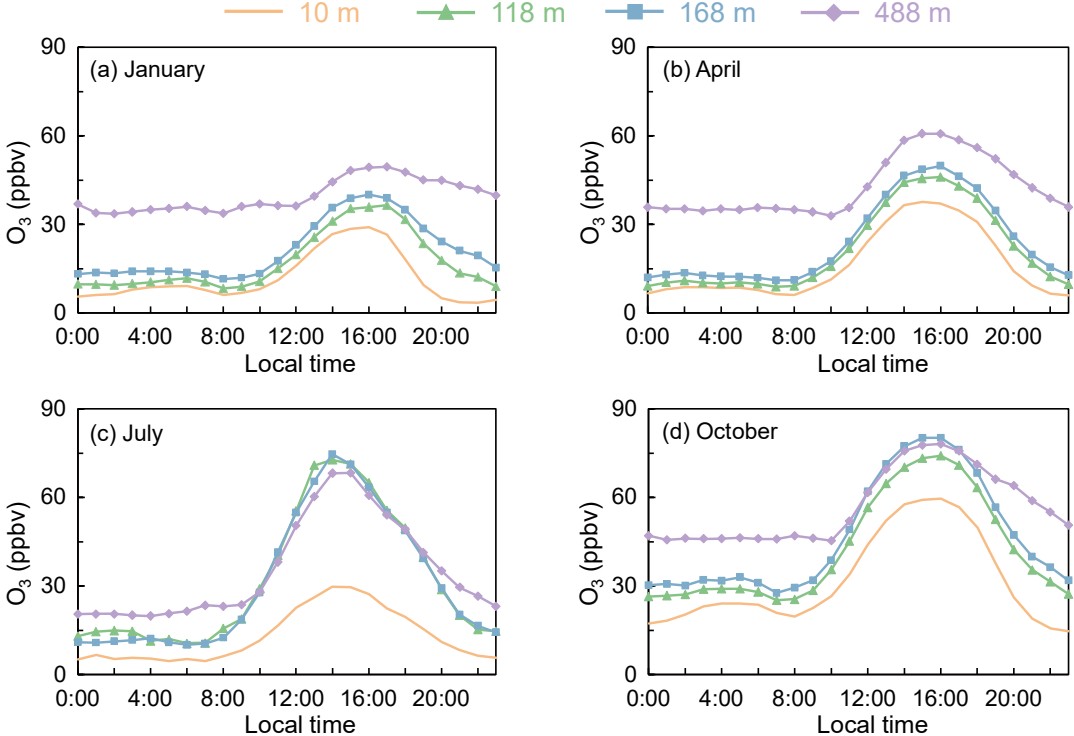

**Figure 2.** Diurnal cycle and seasonal evolution of ozone in the lower boundary layer observed at the Canton Tower. Panels (a), (b), (c), and (d) show measurements in January, April, July (2019 only), and October averaged for 2017-2019. Measurements are available at 10 m, 118 m, 168 m, and 488 m.



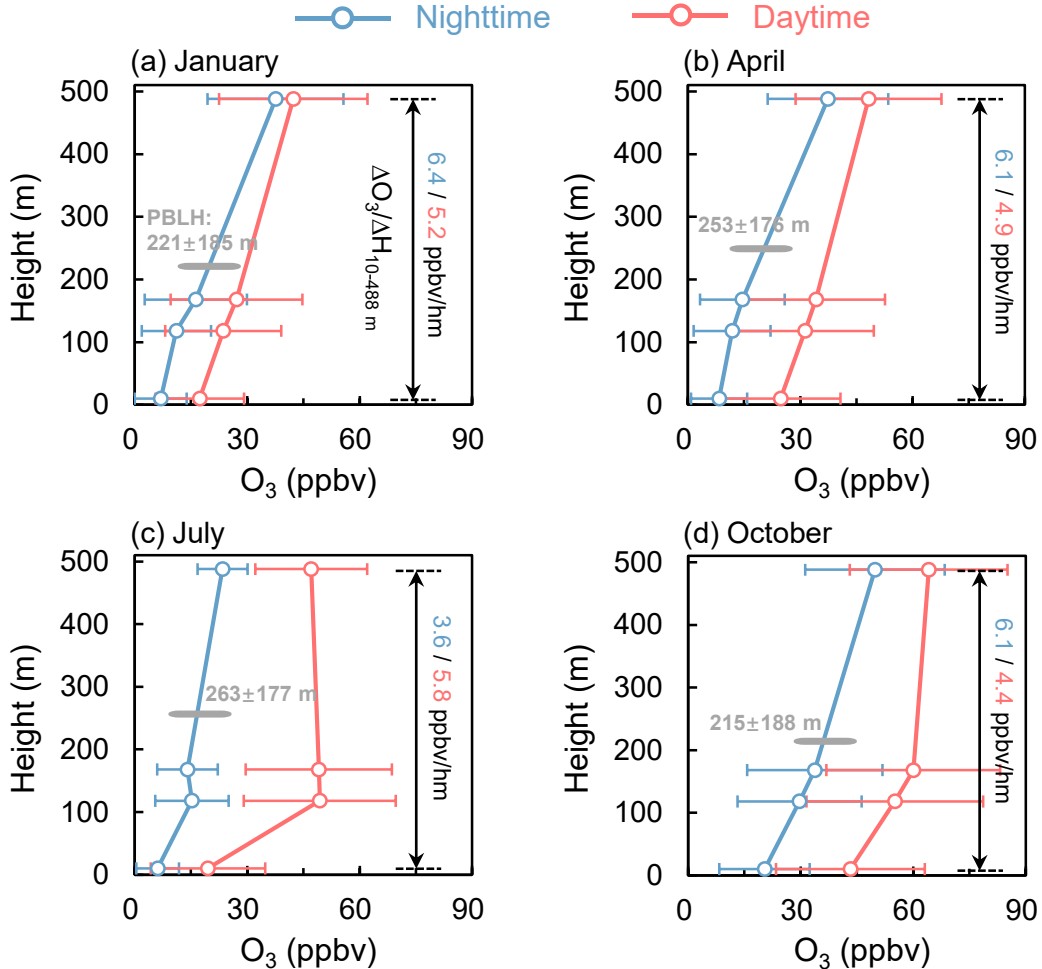

**Figure 3.** 2017-2019 mean ozone vertical structure in the lower boundary layer observed at the Canton Tower. Blue and rosy lines denote nighttime (20:00-07:00 LT) and daytime (08:00-19:00 LT) mean ozone profiles, with the horizontal bars representing standard deviation from the hourly measurements. The horizontal bar and numbers in grey denote the mean height and standard deviation of nighttime stable boundary layer obtained from the ERA5 reanalysis data. $\Delta O_3/\Delta H_{10\text{-}488\ m}$ represents the vertical gradient of ozone between 10 m and 488 m.





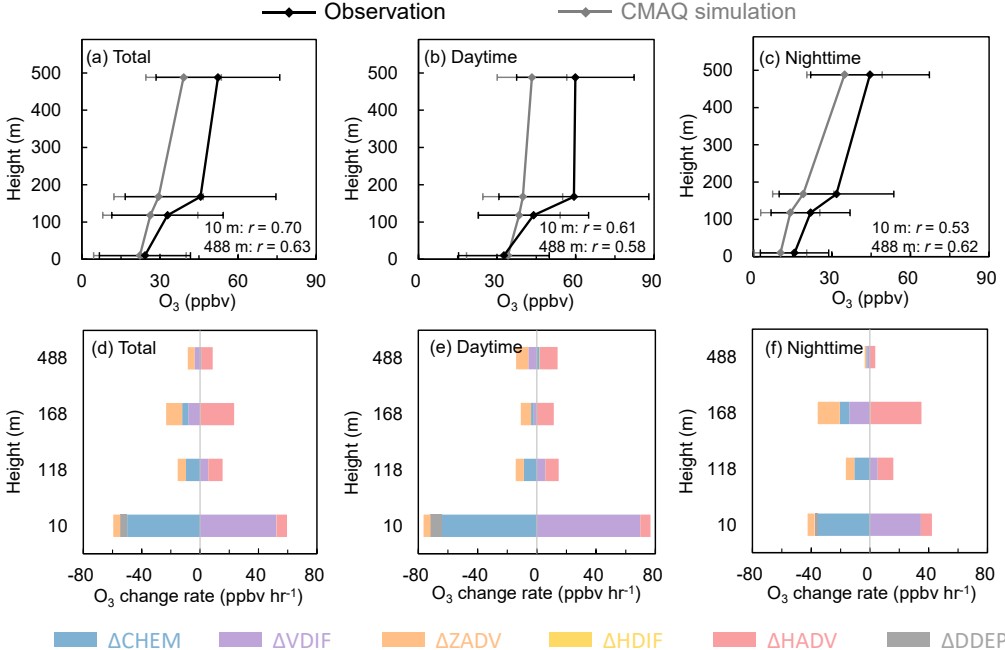

**Figure 4.** CMAQ model simulation of ozone structures and budgets at the Canton Tower, October 2017. Panels (a), (b), and (c) show the mean profile of observation and CMAQ simulation. *r* represents correlation coefficient between hourly observation and simulation at 10 m and 488 m. Horizontal bars represent the standard deviation. Panels (d), (e), and (f) show the ozone budget terms diagnosed from the CMAQ IPR module at different measurement height. ΔCHEM represents change in chemistry, ΔVDIF represents change in vertical diffusion, ΔZADV represents change in vertical advection, ΔHDIF represents change in horizontal diffusion, ΔHADV represents change in horizontal advection, and ΔDDEP represents change in dry deposition.





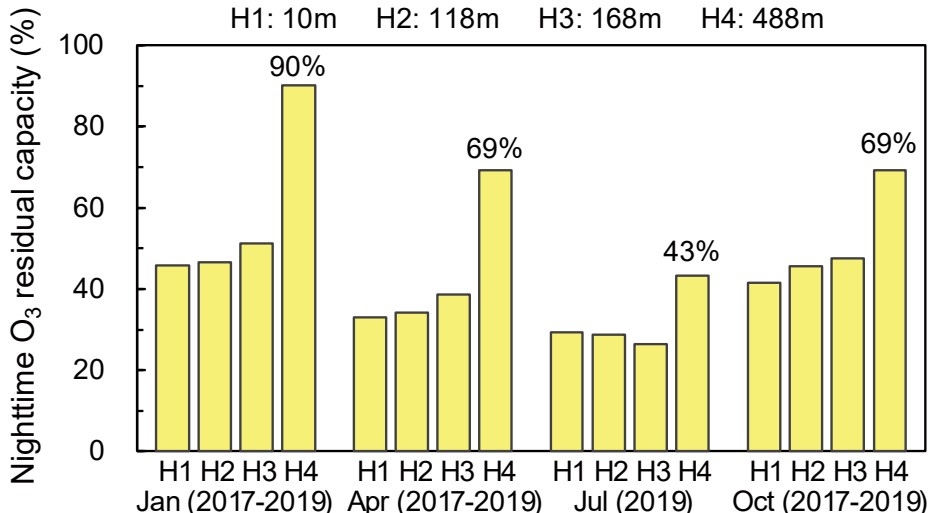

**Figure 5.** Nighttime ozone residual capacity at different height in the lower boundary layer in four seasons obtained from the Canton Tower measurements, 2017-2019. The nighttime ozone residual capacity is defined as the ratio of the ozone concentrations averaged over nighttime (20:00-07:00 LT) to that averaged over afternoon (14:00-17:00 LT).



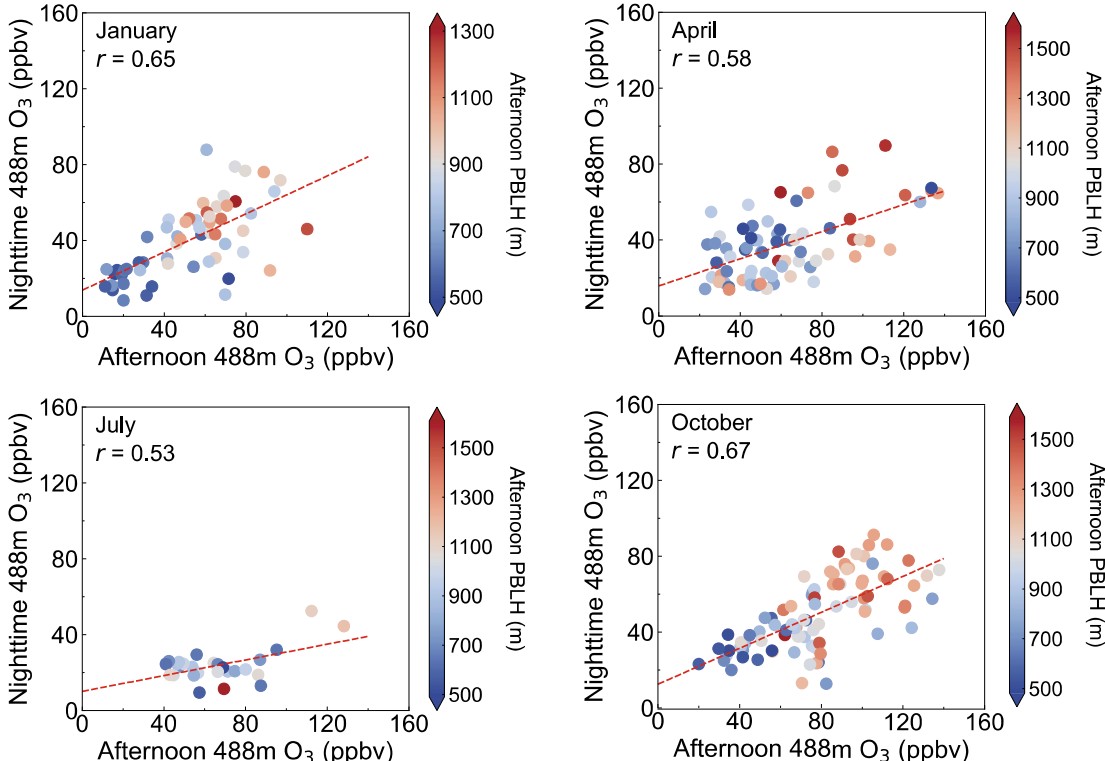

**Figure 6.** Dependence of nighttime (20:00-07:00 LT) 488 m ozone concentrations on afternoon (14:00-17:00 LT) 488 m ozone
concentrations. $r$ represents correlation coefficient between the two variables. The red dash line is fit from the two variables
by a first-order polynomial. The color represents the planet boundary layer height (PBLH) obtained from the ERA5 reanalysis
data averaged over afternoon.





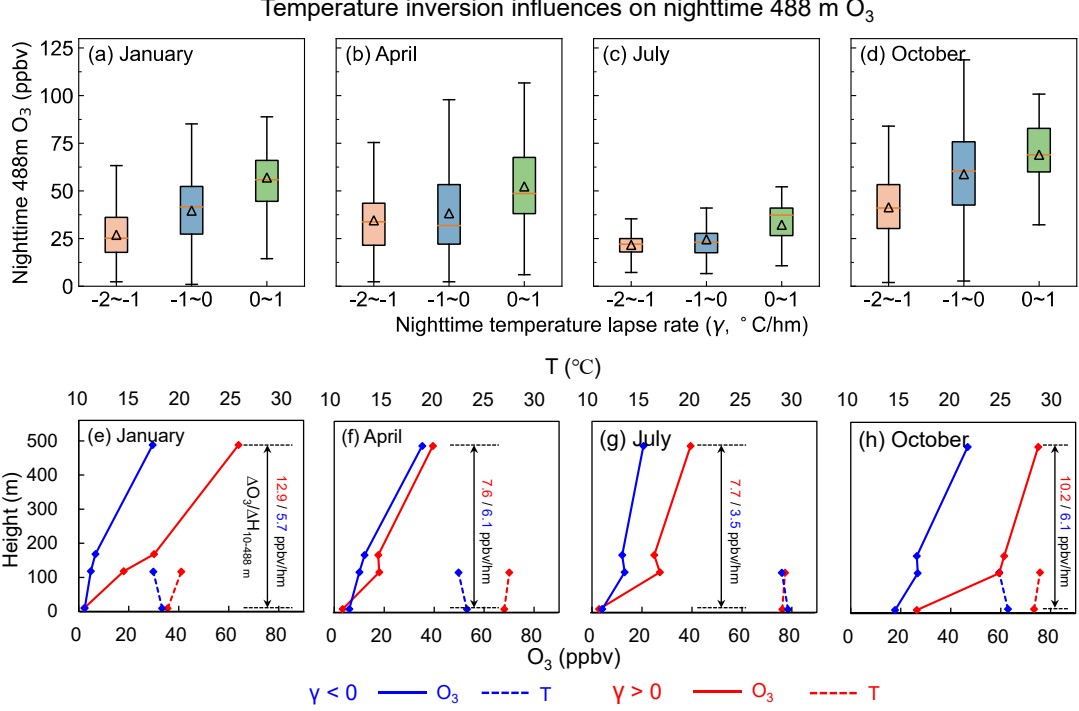

**Figure 7.** Dependence of nighttime 488 m ozone concentrations on temperature inversion, categorized by temperature lapse
rate γ. Panels (a), (b), (c), and (d) show the box-and-whisker plots (minimum, 25th, 50th, 75th percentiles, and maximum) of
nighttime 488 m ozone concentrations on different ranges of γ. Panels (e), (f), (g), and (h) show the nighttime vertical ozone
(bottom x-axis) and temperature profile (top y-axis) with γ < 0 (blue colored lines) versus that with γ > 0 (red colored lines).
$\Delta O_3/\Delta H_{10\text{-}488\ m}$ represents the vertical gradient of ozone between 10 m and 488 m.



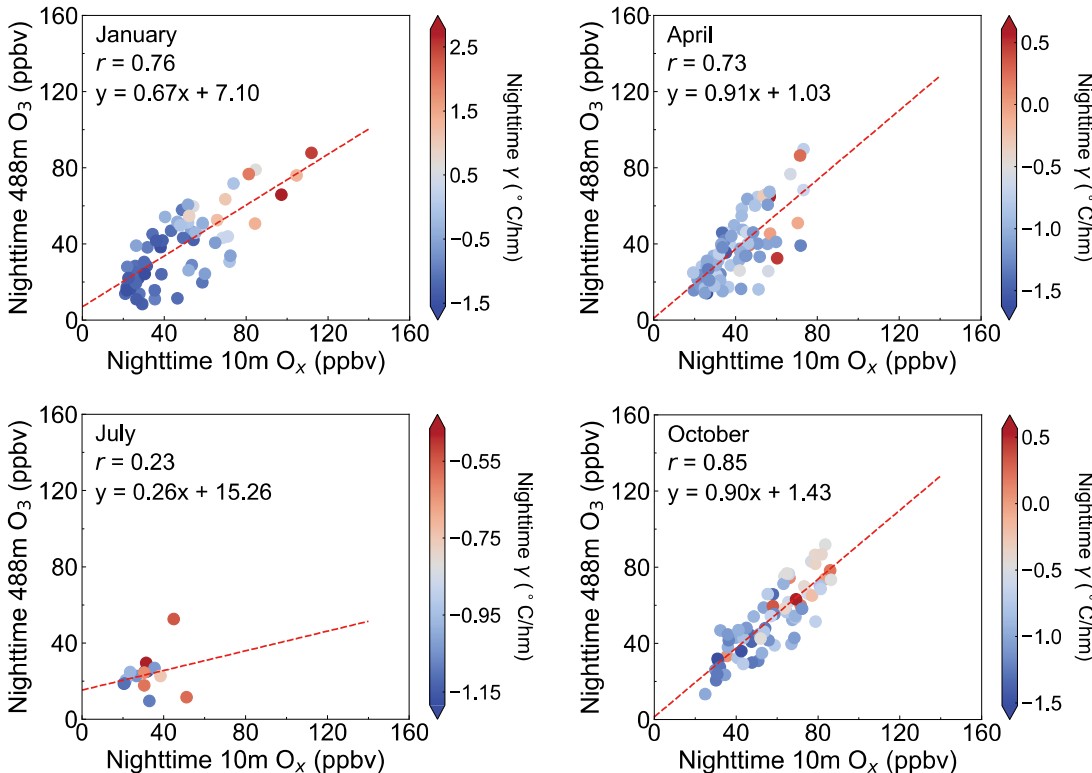

**Figure 8.** Relationship between nighttime (20:00-07:00 LT) 488 m ozone concentrations and nighttime surface $O_x$ ($O_x = O_3 +$ $NO_2$) concentrations. $r$ represents correlation coefficient between the two variables. The red dash line and the fitting formula are fit from the two variables by a first-order polynomial. The fill color represents the temperature lapse rate averaged over nighttime.



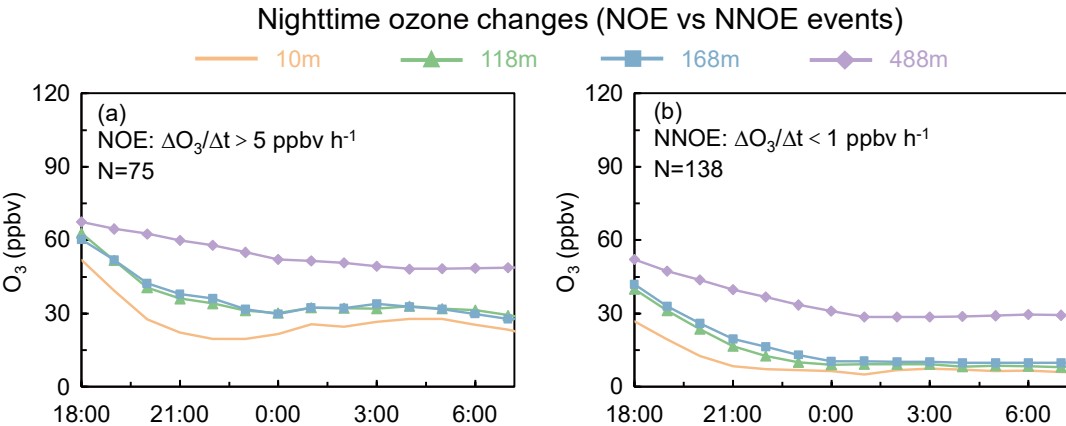

**Figure 9.** Comparison of ozone time series between the nighttime ozone enhancement (NOE, maximum of $\Delta O_3/\Delta t > 5$ ppbv h$^{-1}$) and the non-enhanced nocturnal ozone (NNOE, maximum of $\Delta O_3/\Delta t < 1$ ppbv h$^{-1}$) events. Panels (a) and (b) show the nighttime ozone changes averaged over 75 NOE and 138 NNOE events in 2017-2019 with available observations, respectively.





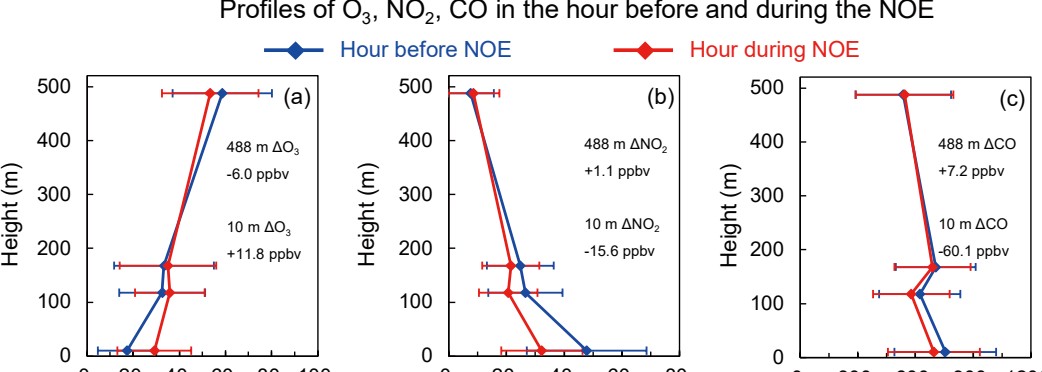

**Figure 10.** Characteristics of ozone, NO₂, CO profiles before (blue colored lines) and during (red colored lines) the occurrence
of the NOE event. ΔO₃, ΔNO, and ΔCO represent the change of concentrations during the occurrence of the NOE event
compared to the hour before the NOE event.



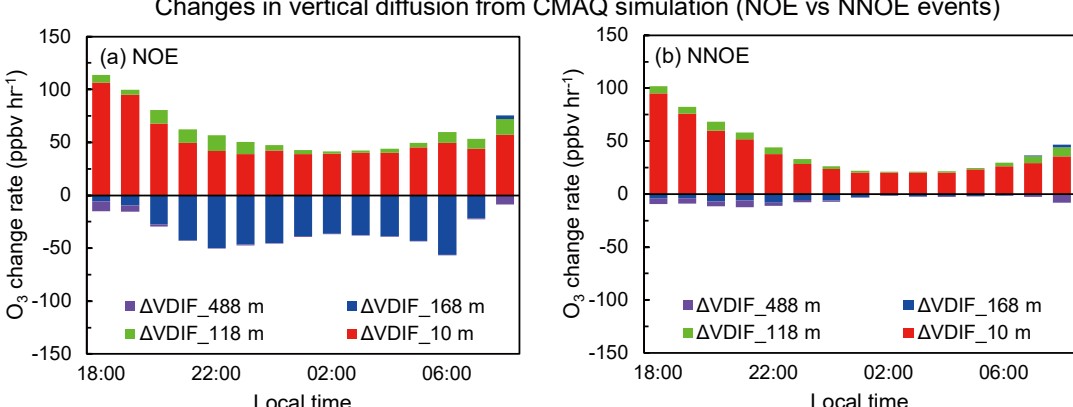

**Figure 11.** Characteristics of vertical diffusion (ΔVDIF) during NOE and NNOE events at the four measurement heights at the Canton Tower. The budget terms of ΔVDIF are diagnosed from the CMAQ IPR module budget at the Canton Tower, October 2017.



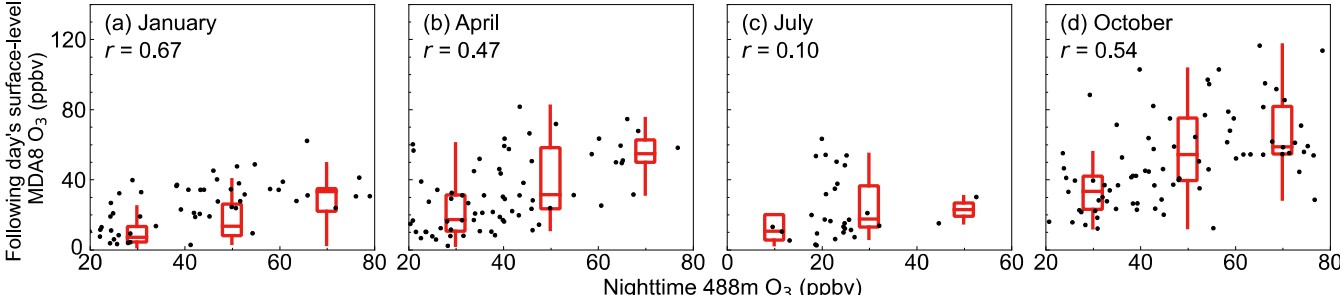

**Figure 12.** Relationship between nighttime 488 m ozone and the following day's surface-level MDA8 ozone. *r* represents correlation coefficient between the two variables. The box-and-whisker plots (minimum, 25th, 50th, 75th percentiles, and maximum) represent the dependence of following day's surface-level MDA8 ozone on different range of nighttime 488 m ozone.





**Figure 13.** CMAQ simulated ozone changes in the early morning, October 2017. Panel (a) shows the simulated diurnal variation of ozone from the surface to 2000 m at the Canton Tower. Numbers denote the vertical diffusion (ΔVDIF) between 08:00 and 09:00 LT at the surface (10 m) and the 488 m height, diagnosed by the IPR module. Panels (b), (c), and (d) show the 488 m ozone concentrations at 08:00 and 09:00 LT and their difference. Panels (e), (f) and (g) are the same as panels (b), (c), and (d) but for the 10 m height. The triangle marks the location of the Canton Tower.





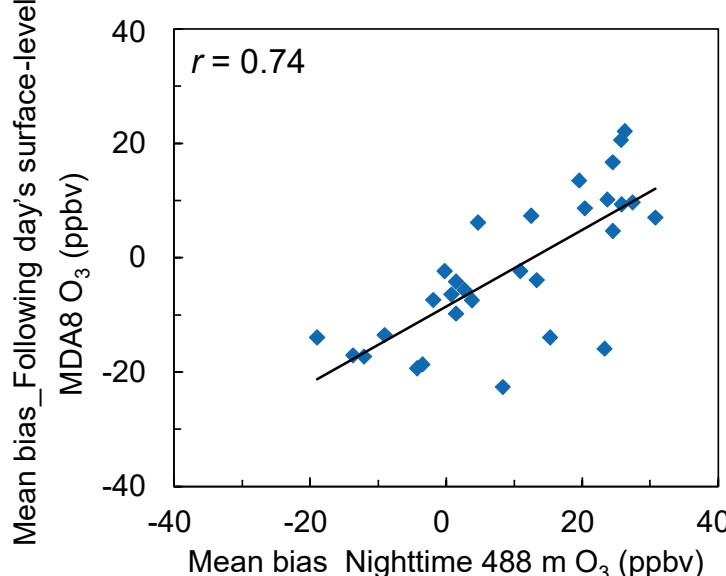

**Figure 14.** Relationship between the CMAQ model bias in nighttime ozone at the 488 m height and bias in the following day's surface MDA8 ozone. *r* represents correlation coefficient between the two variables.



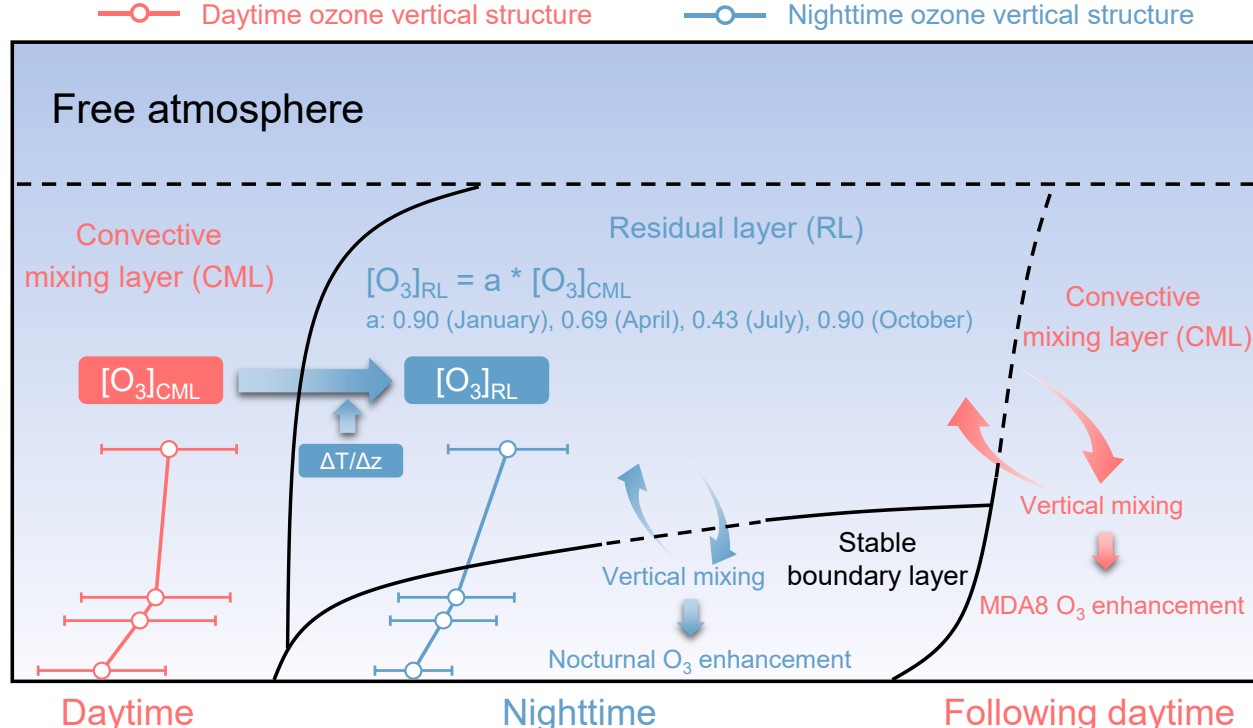

**Figure 15.** Conceptual model of the nighttime ozone in the lower boundary layer and its influences on surface ozone. The ozone profiles and the residual capacity (a) are obtained from the measurements in 2017-2019.



**Table 1.** Summary of key configurations in WRF-CMAQ modeling system

|  | Configuration |
| --- | --- |
| Microphysics | Purdue Lin (Chen and Sun, 2002) |
| PBL physics scheme | MYJ (Janji, 1994) |
| Shortwave and longwave radiation | Goddard and Rapid Radiative Transfer Model (RRTM) (Mlawer et al., 1997) |
| Land surface model | Noah land surface model (LSM) (Ek et al., 2003) |
| Urban scheme | Single-layer urban canopy model (UCM) (Kusaka and Kimura, 2004) |
| Gas-phase chemistry | CB06 (Yarwood, 2010) |
| Aerosol module | AERO6 (Murphy et al., 2017; Pye et al., 2017) |
| Meteorological initial conditions and boundary conditions | NECP Final (FNL) reanalysis data |
| Anthropogenic emissions | MEIC in 2017 (Zheng et al., 2018) |
| Biogenic emissions | MEGANv2.1 (Guenther et al., 2012) |



**Table 2.** Summary of nighttime versus daytime ozone at the highest (488 m) and lowest (10 m) measurement platform at the Canton Tower

| Month | Layer (m) | Ozone (ppbv) | | 488 m ozone/10 m ozone | |
|---|---|---|---|---|---|
| | | Nighttime | Daytime | Nighttime | Daytime |
| January | 10 | 7±7 | 17±12 | 5.4 | 2.4 |
| | 488 | 38±18 | 42±20 | | |
| April | 10 | 8±7 | 25±16 | 4.5 | 1.9 |
| | 488 | 37±16 | 48±19 | | |
| July | 10 | 6±6 | 20±15 | 3.8 | 2.4 |
| | 488 | 23±7 | 47±15 | | |
| October | 10 | 20±12 | 43±20 | 2.4 | 1.5 |
| | 488 | 50±19 | 64±21 | | |



**Table 3.** Summary of nighttime versus daytime monthly-mean vertical gradient of ozone

| $\Delta H$ (hm) | Period | $\Delta O_3/\Delta H$ (ppbv/hm) | | | |
|---|---|---|---|---|---|
| | | January | April | July | October |
| 488-10 | Nighttime | 6.4 | 6.1 | 3.6 | 6.1 |
| | Daytime | 5.2 | 4.9 | 5.8 | 4.4 |
| 118-10 | Nighttime | 3.9 | 3.2 | 8.5 | 8.6 |
| | Daytime | 5.8 | 6.0 | 27.6 | 11.0 |
| 488-168 | Nighttime | 6.6 | 7.1 | 2.9 | 5.0 |
| | Daytime | 4.7 | 4.4 | -0.6 | 1.3 |