# Peer review of "Nighttime ozone in the lower boundary layer: insights from 3-year tower-based measurements in South China and regional air quality modeling"

_EGUsphere, 2023_

## Author Comment (AC1)

**Reviewer #1**

**Comment [1-1]:** General comments: In this manuscript, the authors gave a very details analysis mostly based on 3 year ozone and other gas pollutants observations on a 488m high Canton Tower with 4 levels. The data collected were precious, and the topic is of great interesting to recognize ozone vertical exchanges within boundary layer (BL) and related to ozone diurnal variation. The analysis is mostly sound, but some details need clarify.

**Response [1-1]: We thank the reviewer for the positive and valuable comments. All of them have been implemented in the revised manuscript. Please see our itemized responses below.**

**Comment [1-2]:** Specific comments: Refer to the discussion in 341-344, the study address the cases that rule out wind speeds above 2 m/s at four vertical layers, and say "exclude the possible influence of horizontal transport". My concern is that in a consecutive event (NOE), some periods could be in smaller wind speed, sometimes in bigger. While, I do not think that means horizontal transport is not important in a NOE. My interesting is what about the influence of horizontal transport, because horizontal transport from rural sites may also induce high surface ozone and the emission from high stacks could also contribute to high CO and $NO_2$ in RL. I suggest a detailed analysis of some typical cases, also at least compared the model results of $\Delta VDIF$ and $\Delta ADV$ in different levels.

**Response [1-2]: Thank you for your suggestions. We agree that horizontal transport from rural sites have the potential to elevate surface ozone in nighttime. We have added the statement in Section 3.3 "Ozone concentrations are higher in nighttime RL than at the surface, while $NO_2$ and CO concentrations are much higher at the surface as they are primarily released from anthropogenic emissions at near surface while there is almost no direct source at higher altitudes near the Canton Tower. As such, the increase in $NO_2$ and CO at 488 m during the NOE events, even though with very small magnitude, is most likely from the vertical mixing with surface layer air mass or from horizontal transport of polluted air parcels". We have also followed your suggestion to conduct two cases study to illustrate the contribution of horizontal transport on NOE event in Section 3.3. Please kindly refer to Response [3-6] for more details on the case studies.**

**Comment [1-3]:** This study identify a strong ozone residual capacity, defined as the ratio of the ozone concentration averaged over nighttime to that in the afternoon (14:00-17:00 LT). As the ratio in mathematics, I suggest to replace the definition of "ozone residual capacity" to "ozone residual ratio".

**Response [1-3]: We agree. We have replaced the definition of "ozone residual capacity" to "ozone residual ratio" in both Figure 5 and relevant contexts.**

**Comment [1-4]:** It's not exact to say "weakens the titration effect by diluting $NO_x$ concentrations." during NOE. It could be better to say "offset the surface ozone

decrease by NO diluting".

**Response [1-4]: Thank you for pointing it out. We have revised the text for clarification: "The enhanced vertical mixing leads to NOE event by introducing ozone-rich and NO$_x$-poor air in the RL to enter the nighttime stable boundary layer."**

**Comment [1-5]:** The explanation in line 270-279 were not sufficient. Except the "ozone-rich air at higher altitudes to mix with air in the lower boundary layer.", the higher ozone produced by photochemistry near surface should also convectively transport to upper BL or low free troposphere.

**Response [1-5]: Thank you for correction. We have added the following text in paragraph 3 of Section 3.2 as follows for justification: "allowing ozone-rich air from higher altitudes and ozone chemically produced near surface to mix with air in the lower boundary layer."**

**Comment [1-6]:** In fig. 12, what's the result if you statics the relation of surface ozone before noon (for example 9:00-11:00) and before the sunrise (for example 5:00-6:00)?

**Response [1-6]: We further analyze the relationship between surface ozone before noon (averaged over 9:00-11:00) and before sunrise (averaged over 5:00-6:00), as depicted in Figure R1. We also find a comparable dependency of the two variables, consistent with Figure 12.**

[Figure]

**Figure R1. Relationship between nighttime 488 m ozone before sunrise (averaged over 5:00-6:00) and the following day's surface-level ozone before noon (averaged over 9:00-11:00).**

[Figure]

**Figure 12. Relationship between nighttime 488 m ozone and the following day's surface-level MDA8 ozone.**

**Comment [1-7]:** In introduction, in line 58-61, I suggest move and combined these sentences that introduce your study to line around 95. Also, please polish the context

and concisely present the analysis.

**Response [1-7]: Thank you for pointing it out. In order to avoid redundancy in introduction in line 58-61, we have revised as: "Here we combine observations and model simulation to analyze nighttime ozone in the lower boundary layer in Guangzhou, the core megacity in South China". We prefer to have a short sentence to outline the major content of our study in the early part of the introduction.**

**We have polished the context in multiple places in the manuscript following your suggestion.**

---

## Author Comment (AC2)

**Reviewer #2**

**Comment [2-1]:** General comments: The paper by He et al., investigated vertical distributions and process contributions of the nighttime boundary-layer ozone in Southern China using 3-year tower-based measurements. As indicated by the authors, the continuous gradient measurements of ozone in the lower boundary layer, particularly in urban regions, are very important for clarifying vertical exchange characteristics of ozone and thus elucidating the reasons that regulate surface ozone air quality. The paper is well written and organized. The analysis regarding vertical distributions and key drivers of ozone was reasonable and has been well supported in the literature. Therefore, I only have some small concerns that may be further explained by the authors before its publication.

**Response [2-1]: We thank the reviewer for the positive and valuable comments. All of them have been implemented in the revised manuscript. Please see our itemized responses below.**

**Comment [2-2]:** Specific comments: Line 115: How the wind information was measured on the tower? was it measured inside or outside the outer shell of tower? The structure of the tower may cause complicated and different turbulences affecting the measured winds on different altitudes.

**Response [2-2]: Thank you for pointing it out. The wind speed at the lower altitude is obtained from an observation station beside the Canton Tower. The wind speed at the middle layers (118 m, 168 m) is measured inside the outer shell of the tower. The wind speed at the highest layer (488 m) is measured inside the hollow mast at the tip of the tower.**

**We agree that the structure of the tower may affect the measured winds on different altitudes. In our study, we find that the dependence of the nighttime 488 m ozone concentration on temperature vertical lapse rate does not show obvious difference under various wind speed condition over the 3-year measurements. According to the previous studies on measurements from the Canton Tower, we have added the following text within paragraph 2 of Section 2.1 as follows: "The structure of the tower may cause complicated and different turbulences affecting the measured winds on different altitudes, however, the above previous studies have demonstrated the reliability of atmosphere components measured from the Canton Tower."**

**Comment [2-3]:** Line 205: The vertical gradients of ozone in the nighttime boundary were much stronger than in the daytime due to the inhibition of the vertical mixing. The authors also stated that the vertical gradients of $O_x$ mixing ratios are much weaker than those of ozone. Therefore, the positive gradients of vertical ozone profiles were mainly determined by the gradually reduced NO titration effect. Were any vertical gradient measurements of NO on the tower that can be used to support this conclusion?

**Response [2-3]: Thank you for pointing it out. Following your suggestion, we have**

added **Figure S3 to show the vertical gradient of measured NO$_x$. We find that vertical gradient measurements of NO$_x$ can be used to support the vertical gradients of ozone between nighttime and daytime. We have added the following text in Section 3.1 as follows: "We also find a larger vertical gradient of NO$_x$ (NO+NO$_2$) concentrations in nighttime than daytime (Fig. S3), suggesting that the titration effect is an important factor shaping the ozone vertical structure."**

[Figure]

**Figure S3. Mean NO$_x$ (NO+NO$_2$) vertical structure in the lower boundary layer observed at the Canton Tower, October 2017. Blue and rosy lines denote nighttime (20:00-07:00 LT) and daytime (08:00-19:00 LT) mean NO$_x$ profiles, respectively.**

**Comment [2-4]:** Line 235-240: As highlighted by the authors, the discrepancies between modeled and measured ozone in urban regions aloft may be caused by many factors. In my opinion, a grid with a small spatial scale of 3×3 km$^2$ in the urban region may be not the dominant factor causing these significant differences. I suggest that the authors can provide a comparison between the modeled and measured vertical profiles of NO$_x$ in the boundary layer to check whether exist significant discrepancies. In addition to the state of vertical mixing, the vertical distribution of NO$_x$ is also an important factor to shape the vertical profile of ozone in the boundary layer. Furthermore, as reported in the literature, ambient NO$_x$ concentrations declined rapidly in recent years in China and thus I am not sure whether the emission inventory of NO$_x$ used in the model can well reflect these changes.

**Response [2-4]: Thank you for your suggestions. The MEIC inventory used in this study provides anthropogenic emission in China for year 2017, which covers the simulation period of this study.**

**We compare the mean vertical profiles of NO$_x$ in the lower boundary layer between the model simulation and measurement, as shown in Figure R2. The model mostly captures the observed magnitude and vertical structure of NO$_x$ both in daytime and nighttime, but still has notable bias at the surface, suggesting uncertainties in emission inventory. We have added the following text in Section 2.3 to clarify the use of anthropogenic emission inventory in this study "Anthropogenic emissions from MEIC in 2017 are downscaled from its original resolution of 25×25 km$^2$ to the model resolution of 3×3 km$^2$, using the Modular**

**Emission Inventory Allocation Tool for Community Multiscale Air Quality Model (MEIAT-CMAQ) v1.0 (Wang et al., 2023a)", and in Section 3.1 for the model bias discussion "The lack of high-resolution (e.g. ~3km or higher) anthropogenic emission inventory may cause bias for simulation of ozone precursors."**

[Figure]

**Figure R2. Mean NO$_x$ vertical structure in the lower boundary layer of observation and CMAQ simulation.**

**Reference:**

Wang, H., Qiu, J., Liu, Y., Fan, Q., Lu, X., Zhang, Y., Wu, K., Shen, A., Xu, Y., Jin, Y., Zhu, Y., Sun, J., and Wang, H.: MEIAT-CMAQ v1.0: A Modular Emission Inventory Allocation Tool for Community Multiscale Air Quality Model Version 1.0, EGUsphere, 2023, 1-33, https://doi.org/10.5194/egusphere-2023-1309, 2023a.

**Comment [2-5]:** Line 245-250 and Figure 4: These results and conclusions are quite confusing. In nighttime, process contributions of the change in ozone at different altitudes are plausible. However, process contributions of the change in ozone in daytime are confusing. As shown in Figure 4, the increase in surface ozone in daytime were mainly contributed by vertical diffusion and horizontal advection. The authors also highlight that chemistry is not a major source/sink of ozone at 488 m. According to these results, the boundary-layer ozone budget in urban Guangzhou was mainly contributed by transport from adjacent regions or from even higher altitudes? Local formation of ozone from photochemistry has negligible contributions to the increase in the boundary layer ozone in daytime?

**Response [2-5]: Thank you for pointing it out. Our results of ozone budget diagnostics are consistent with previous study conducted in Hong Kong (Wang et al., 2015) and Guangzhou (Xu et al., 2023) using CMAQ model, all showing negative contributions of photochemistry to surface ozone concentrations and positive contribution at higher altitudes in urban area. This may reflect strong chemical loss by high NO$_x$ at the surface once the ozone is produced. Here we additionally present the ozone vertical budget extending from surface to about 2 km ahead at noon when the ozone chemical production is the most intense, as depicted in Figure R3. We indeed find significant net ozone production in the boundary layer that contribute to ozone increase, and these ozone enhancements can be transported and/or diffused to surface level. We have added the following**

text in Section 3.1 as follows: **"In noon when chemical production is intense, however, we find that the ΔCHEM exhibits positive contribution at ~200-1000 m, while ΔVDIF exhibits negative contribution at ~200-1000 m but positive at the surface. These budgets are consistent with the previous study conducted in Hong Kong (Wang et al., 2015) and Guangzhou (Xu et al., 2023), indicating that surface ozone is mainly contributed by vertical diffusion from local photochemistry in higher altitudes in urban Guangzhou."**

[Figure]

**Figure R3. CMAQ model simulation of ozone budgets at the Canton Tower, October 2017. ΔCHEM represents change in chemistry, ΔVDIF represents change in vertical diffusion, ΔZADV represents change in vertical advection, ΔHDIF represents change in horizontal diffusion, ΔHADV represents change in horizontal advection, and ΔDDEP represents change in dry deposition.**

**Reference:**

Wang, N., Guo, H., Jiang, F., Ling, Z. H., and Wang, T.: Simulation of ozone formation at different elevations in mountainous area of Hong Kong using WRF-CMAQ model, Sci. Total Environ., 505, 939-951, https://doi.org/10.1016/j.scitotenv.2014.10.070, 2015.

Xu, Y. F., Shen, A., Jin, Y. B., Liu, Y. M., Lu, X., Fan, S. J., Hong, Y. Y., and Fan, Q.: A quantitative assessment and process analysis of the contribution from meteorological conditions in an O$_3$ pollution episode in Guangzhou, China, Atmos. Environ., 303, https://doi.org/10.1016/j.atmosenv.2023.119757, 2023.

**Comment [2-6]:** Line 260-265: In daytime, the enhancement of air turbulence could drive the well mixing of ozone in the boundary layer. Therefore, the defined "nighttime ozone residual capacity" may be not suitable to assess the influence of the ozone at a certain height in the nighttime on the ozone budget at the same height in the daytime boundary layer.

**Response [2-6]:** We agree, thank you for correction. The defined "nighttime ozone

residual capacity" ignore the contribution from horizontal transportation and vertical mixing to ozone at a certain height. It only reflects the relative amount of ozone concentrations averaged over nighttime to that averaged over afternoon.

To prevent potential misleading to readers, we have replaced the definition of "nighttime ozone residual capacity" to "nighttime ozone residual ratio" in a mathematically manner. Specifically, we have made revisions to the statement in Section 3.2 as follows: "We note that this ratio at a certain height only quantifies the averaged level of nighttime ozone compared to afternoon when ozone concentrations typically reach the daily peak, and does not account for additional influence of horizontal transportation and vertical mixing. However, it can still serve as a useful metric to quantify to what extent ozone in the afternoon can be reserved in the nighttime."

**Comment [2-7]:** Line 340-344: How $O_x$ changes at different altitudes during the NOE events?

**Response [2-7]: During the NOE event, the observed $O_x$ concentration becomes more consistent between the surface and the higher altitudes, as shown in Figure R4. It is a signal for the enhanced vertical mixing.**

[Figure]

**Figure R4. Characteristics of $O_x$ profiles before (blue colored lines) and during (red colored lines) the occurrence of the NOE event.**

**Comment [2-8]:** Line 370-375: I agree with the authors' opinion that the improvement of the model capability in simulating nighttime ozone in the RL is a key to decreasing errors of the modeling results. The timely update of the $NO_x$ emission inventory may be another important approach to improve the accuracy of the ozone modeling results.

**Response [2-8]: We agree. We have reflected this point in Section 3.1. Please kindly refer to Response [2-4].**

---

## Author Comment (AC3)

**Reviewer #3**

**Comment [3-1]:** General comments: Nighttime ozone in the lower boundary layer and its influences on surface ozone: insights from 3-year tower-based measurements in South China and regional air quality modeling. This is an interesting manuscript on vertical structure of ozone in the lower atmosphere based on 3-year tower-based measurements in South China and air quality model analysis. The manuscript is clearly written and well organized.

**Response [3-1]: We thank the reviewer for the positive and valuable comments. All of them have been implemented in the revised manuscript. Please see our itemized responses below.**

**Comment [3-2]:** Specific comments: The title is confusing a little bit. "Nighttime ozone in the lower boundary layer" includes surface ozone, right? How to understand "its influences"?

**Response [3-2]: Thank you for pointing it out. We have revised the title as follows: "Nighttime ozone in the lower boundary layer: insights from 3-year tower-based measurements in South China and regional air quality modeling"**

**Comment [3-3]:** How many nighttime ozone enhancement (NOE) events occurred in the 3-year tower-based measurements? Does NOE happen at the same day in other air quality stations in Guangzhou? Are there seasonal differences in NOE? Authors stated that "During the surface nighttime…a typical feature of enhanced vertical mixing" (lines 32-34). Is there another way leading to NOE in Guangzhou, for example, horizontal advection. A period of 3 years is not short.

**Response [3-3]: In our study, measurements are available in 2017-2019 for January, April, and October, but are only available in 2019 for July due to instrument malfunction in other summers. We find 75 NOE events which account for 24% among the available measurements.**

**We follow previous studies to define an NOE event if surface ozone concentration increases by more than 5 ppbv ($\Delta O_3/\Delta t > 5$ ppbv h$^{-1}$) in one of any two adjacent hours during nighttime. Accordingly, we analyze the measurements collected from 4 air quality stations near the Canton Tower. The results reveal that NOE events occur simultaneously in the Canton Tower and in other air quality stations in Guangzhou for 67% of the NOE episodes.**

**The frequency of NOE event exhibits seasonal differences, we have added the following text in Section 3.3 as follows: "The frequency of NOE events follow the seasonal pattern of Autumn (37%) > Winter (32%) > Spring (11%) > Summer (3%), which consistent with Wu et al. (2023) for the period 2006-2019 in the Pearl River Delta region, which also shows a higher frequency in Autumn and a lower frequency in Summer."**

We have followed your suggestion to conduct two cases study and examine the factors leading to NOE in Guangzhou. We find the vertical mixing is the major impact factor for surface ozone enhancement in case I, while vertical mixing and horizontal advection contribute equally in case II. Please kindly refer to Response [3-6].

**Reference:**
Wu, Y. K., Chen, W. H., You, Y. C., Xie, Q. Q., Jia, S. G., and Wang, X. M.: Quantitative impacts of vertical transport on the long-term trend of no cturnal ozone increase over the Pearl River Delta region during 2006-2019, Atmos. Chem. Phys., 23, 453-469, https://doi.org/10.5194/acp-23-453-2023, 2023.

**Comment [3-4]:** "oxidation capacity" is not clearly defined in the abstract. How to understand "This indicates a persistent high ozone level and oxidation capacity aloft the surface" (line 23)?

**Response [3-4]: Thank you for pointing it out. To avoid confusion, we have removed the sentence "This indicates a persistent high ozone level and oxidation capacity aloft the surface" from both the Abstract and Conclusions sections.**

**Comment [3-5]:** What is the weather condition favoring "significant influences of nocturnal RL ozone on both nighttime and the following day's daytime surface ozone air quality"?

**Response [3-5]: We have introduced the weather condition favoring "significant influences of nocturnal RL ozone on both nighttime and the following day's daytime surface ozone air quality" in Introduction section and paragraph 2 of Section 3.3 (e.g. "In nighttime, the ozone-rich air in the RL may mix down to surface and trigger a nocturnal ozone enhancement (NOE) event in favorable weather conditions such as the nocturnal low-level jets (Sullivan et al., 2017; He et al., 2022a; Wu et al., 2023)". However, due to word limitation, we do not expand it in detail in the abstract.**

**Reference:**
He, C., Lu, X., Wang, H. L., Wang, H. C., Li, Y., He, G. W., He, Y. P., Wan g, Y. R., Zhang, Y. L., Liu, Y. M., Fan, Q., and Fan, S. J.: The unexpect ed high frequency of nocturnal surface ozone enhancement events over Ch ina: characteristics and mechanisms, Atmos. Chem. Phys., 22, 15243-15261, https://doi.org/10.5194/acp-22-15243-2022, 2022a.
Sullivan, J. T., Rabenhorst, S. D., Dreessen, J., McGee, T. J., Delgado, R., Tw igg, L., and Sumnicht, G.: Lidar observations revealing transport of $O_3$ in the presence of a nocturnal low-level jet: Regional implications for "next-d ay" pollution, Atmos. Environ., 158, 160-171, https://doi.org/10.1016/j.atmos env.2017.03.039, 2017.
Wu, Y. K., Chen, W. H., You, Y. C., Xie, Q. Q., Jia, S. G., and Wang, X.

M.: Quantitative impacts of vertical transport on the long-term trend of no
cturnal ozone increase over the Pearl River Delta region during 2006-2019,
Atmos. Chem. Phys., 23, 453-469, https://doi.org/10.5194/acp-23-453-2023,
2023.

**Comment [3-6]:** Model performances are evaluated only on monthly bases, while NOE does not happen every day. How does the model capture typical NOE? I suggest two or three typical cases are analyzed in order to see its major impact factors.

**Response [3-6]: Thank you for the suggestion. We follow previous studies to define an NOE event if surface (10 m) ozone concentration increases by more than 5 ppbv ($\Delta O_3/\Delta t > 5$ ppbv h$^{-1}$) in one of any two adjacent hours during nighttime. We identified a total of 10 NOE events based on tower-based measurements during the model simulation period. Out of these 10 events, the model successfully captures 5 cases that matched the defined criteria.**

**We have followed your suggestion to conduct two typical cases and analyze the major impact factors on NOE events. Figure S5 have been added to Supporting Information. The detail analyses in the following text have been added to paragraph 6 of Section 3.3 as follows:** **"We zoom in the processes leading to NOE events in two episodes, October 24 (case I) and October 28-29 (case II) in 2017, as depicted in Figure S5. We quantify the physical and chemical influences on ozone budget at 02:00 LT and 00:00 LT when observed ozone concentration at 10 m increases by 17 ppbv and 18 ppbv in case I and case II, respectively. CMAQ model successfully capture the ozone enhancement in both two episodes. At the surface level, the ΔCHEM contributes significantly to ozone destruction, while ΔVDIF and ΔHADV positively contribute to ozone enhancement in both cases. The ΔVDIF is the major impact factor for surface ozone enhancement in case I, while ΔVDIF and ΔHADV contribute equally in case II. We find that in case II, horizontal advection also contributes significantly in the boundary layer. This is associated with the occurrence of a low-level jet, as evident by the high horizontal wind speed exceeding 12 m/s recorded in 950 hPa (from the ERA5 dataset) in the midnight at the location to the Canton Tower (Figure S6). The low-level jet not only brings air with rich ozone concentration from the north, but also enhances vertical mixing by producing turbulent kinetic energy and weakening the decoupling of the RL and the stable boundary layer (He et al., 2022a). This suggests a combined contribution of horizontal transport and vertical diffusion to the NOE event."**

**Reference:**
He, C., Lu, X., Wang, H. L., Wang, H. C., Li, Y., He, G. W., He, Y. P., Wan
g, Y. R., Zhang, Y. L., Liu, Y. M., Fan, Q., and Fan, S. J.: The unexpect
ed high frequency of nocturnal surface ozone enhancement events over Ch
ina: characteristics and mechanisms, Atmos. Chem. Phys., 22, 15243-15261,
https://doi.org/10.5194/acp-22-15243-2022, 2022a.

[Figure]

**Figure S5. CMAQ model simulation of two NOE events. Panels (a) and (b) show the simulated ozone of case I from the surface to 1000 m at the Canton Tower and ozone budget terms diagnosed from the CMAQ IPR module at different measurement height. Panels (c) and (d) are the same as panels (a) and (b) but for case II. ΔCHEM represents change in chemistry, ΔVDIF represents change in vertical diffusion, ΔZADV represents change in vertical advection, ΔHDIF represents change in horizontal diffusion, ΔHADV represents change in horizontal advection, and ΔDDEP represents change in dry deposition.**

[Figure]

**Figure S6. CMAQ model simulation of two NOE events. Panels (a) and (b) show**

the simulated surface ozone and the horizontal and vertical wind from the ERA5 dataset of case I. Panels (c) and (d) are the same as panels (a) and (b) but for case II. The triangle marks the location of the Canton Tower.